# Crystal Plasticity Finite Element Modeling on High Temperature Low Cycle Fatigue of Ti2AlNb Alloy

**Yanju Wang [1]**, **Zhao Zhang [2]**, **Xinhao Wang [2]**, **Yanfeng Yang [2]**, **Xiang Lan [1]** and **Heng Li [2,*]**

[1] Materials Evaluation Center for Aeronautical and Aeroengine Application, AECC Beijing Institute of Aeronautical Materials, Beijing 100095, China

[2] State Key Laboratory of Solidification Processing, Northwestern Polytechnical University, Xi'an 710072, China

[*] Correspondence: liheng@nwpu.edu.cn

**Abstract:** Ti2AlNb alloy is a three-phase alloy, which consists of O phase, $\beta$ phase and $\alpha_2$ phase. Because of the difference in the mechanical characteristics between phases, Ti2AlNb alloy often exhibits deformation heterogeneity. Based on EBSD images of the Ti2AlNb alloy, a crystal plasticity finite element model (CPFEM) was built to study the effects of O phase and $\beta$ phase (dominant phases) on stress and strain distribution. Four types of fatigue experiments, and the Chaboche model with 1.2~1.6% total strain range were conducted to verify the CPFEM. The simulation results showed that the phase boundary was the important position of stress concentration. The main reason for the stress concentration was the inconsistency deformation of grains which resulted from the different deformation abilities of the O and $\beta$ phases.

**Keywords:** Ti$_2$AlNb alloy; low cycle fatigue; Chaboche model; crystal plasticity finite element

## 1. Introduction

Ductile metals fail within about $10^4$ life cycles under periodic plastic loading; this phenomenon is often called the low cycle fatigue (LCF) [1,2]. Aeroengine components are subjected to periodic variations of centrifugal forces and high temperature environments during operation, and the repeated action of high temperature alternating stress results in stress or strain concentration, and then causes local micro plastic deformation. The local micro-plastic deformation causes microcracks to form and spread further in the weak region of the alloy, causing the component to fail in the form of LCF [3]. TiAl-based alloys have received more and more attention because of their excellent high temperature properties and low density [4–6]. In the early of 21st century, many researchers found that adding Nb to TiAl alloys improved high temperature properties and creep resistance, thus, research on TiAlNb alloys has increased in recent years. As an important lightweight alloy, Ti$_2$AlNb alloy is an intermetallic compound. Its long-range ordered super-lattice structure has the effect of weakening dislocation diffusion and high temperature diffusion, which gives the alloy the advantages of high strength, creep resistance, fracture toughness and oxidation resistance [6,7]. Therefore, it is important to study the high temperature LCF of Ti2AlNb alloy by experiments and simulations.

At present, many experimental studies have been conducted on high temperature LCF of TiAlNb alloy, and the fatigue behavior, fatigue life influencing factors, damage evolution and many other aspects have been analyzed. Zhang et al. [8] found that the lamellar O phase of TiAlNb alloy exhibited super fatigue resistance, and fatigue crack growth was characterized mainly by crystallographic cracking. Fang et al. [9] studied the LCF of TiAlNb alloy at 650 °C and ±0.4~±1.2% strain amplitude and determined that the primary $\alpha_2$-phase and $\alpha_2/\beta$ boundaries are important initiation positions for fatigue cracks, and that the lamellar O-phase plays a role in preventing fatigue crack propagation and causes the direction of the microcracks to deviate. Ding et al. [10,11] found that

high temperature LCF will lead to strain-induced phase transformations and dynamic recrystallization. The large difference in lattice strain between different phases results in crack nucleated at and propagated along the phase boundary, which are detrimental to the fatigue life of the TiAlNb alloy. Kruml et al. [12] studied the LCF of TiAlNb alloy at 750 °C, and significant cyclic hardening characteristic was observed for 2%Nb alloy, but cyclic hardening was not observed for 7% Nb alloy, as the higher Nb content increased the stacking fault energy. Fatigue experiment results are lengthy, so establishing models to simulate the LCF process can improve efficiency. At present, the common modeling methods for LCF are macroscopical constitutive models, such as the Chaboche model, and microscopical models, such as the CPFEM.

Macroscopic constitutive models can accurately simulate the stress–strain response of materials, and the few parameters defined in models make them convenient to use. Koo G H et al. [13] constructed a high-temperature viscoplastic cyclic deformation model of 9Cr-1Mo steel based on the Chaboche model to accurately simulate the cyclic elastic–plastic deformation and stress relaxation of the material at 500~600 °C. Vanderson M.D. et al. [14] studied the fatigue of shape memory alloys and proposed a three-dimensional constitutive model to describe the fatigue behavior with a continuum damage perspective. Based on the macroscopic behavior of fatigue crack growth, Chandran [15] established a physical model and constitutive equations to describe the stress life (S-N) fatigue behavior of materials. Ma et al. [16] established a finite element (FE) model with the Chaboche model and ductile damage criterion to simulate the LCF behavior of H11 steel. According to the above research, macroscopic constitutive models based on the Chaboche model can effectively describe the stress–strain response of metal cyclic deformation. However, the macroscopic constitutive method cannot obtain the microstructure of the material during the LCF, and it is difficult to analyze the fatigue mechanism and damage evolution. Therefore, establishing a micro-level model has become a key task.

Among many micro-level models, crystal plastic mechanics is a continuum mechanics method based on crystal dislocation slip theory which can analyze the macroscopic mechanical behavior of materials from the underlying microstructure information. Therefore, many researchers use the crystal plastic method to study the fatigue behavior of materials at grain level. Zhang et al. [17] established a CPFEM to investigate the effects of the microstructure features (grain orientation, grain boundary, and intermetallic compound) on the Sn-rich solder joint fatigue behaviors at −20~400 °C. Han [18] proposed a method to establish polycrystalline CPFEM based on the EBSD measurement of the microstructure and obtained the accumulated plastic strain and slip system diagram. Combined with SEM, EBSD and crystal plastic simulation results, the fatigue deformation mechanism of HEA was comprehensively clarified. Pan et al. [19] predicted the crack nucleation life of coarser crystal (CG) nickel superalloy RR1000 by using CPFEM combined with the critical local storage criterion. Ozturk et al. [20] established a crystal plasticity finite element model to study the effects of microstructure as well as thermal and mechanical loading conditions on fatigue crack nucleation of Ti alloys. Ashton et al. [21] investigated the role of beta phase on room temperature cold dwell fatigue of dual-phase titanium alloys and established a strain gradient crystal plasticity model. The presence of the alpha–beta phase was predicted to increase dwell fatigue resistance compared to a pure alpha phase microstructure. However, the CPFEM of Ti2AlNb alloy high temperature LCF has not been reported on; therefore, it is urgent to establish a CPFEM that can simulate the LCF of Ti2AlNb alloy.

In this paper, a CPFEM was established to conduct the macroscopic stress–strain response and stress and strain distribution at grain level. Firstly, the LCF true stress–strain curves at 650 °C and total strain ranges of 1.2%, 1.3%, 1.4% and 1.6% were obtained. The finite element model of $Ti_2AlNb$ alloy LCF was established by using the Chaboche constitutive model. Then, the CPFEM of O and $\beta$ phases $Ti_2AlNb$ alloy LCF was established, the macroscopic stress–strain responses were obtained and then compared with the results of the experiments and Chaboche model. By analyzing the stress distribution and strain distribution of $\beta$ and O phase, it was found that the phase boundary is the most important

position of stress concentration, and that the main reason for stress concentration is the inconsistet deformation of grains resulting from the different deformation abilities of the O and β phases.

## 2. Experiment and Modeling Theory

### 2.1. Experiment Materials and Procedures

The combustion chamber casing is an annular thin-walled structure. The casing blank is obtained by forging, and then the finished product is obtained by machining the blank. In this paper, $Ti_2AlNb$ alloy is cut from the combustion chamber casing blank. The microstructure of the initial $Ti_2AlNb$ alloy is characterized at room temperature, and the results are shown in Figure 1. The chemical composition of as-received Ti2AlNb alloy is shown in Table 1. According to EBSD (electron backscattered diffraction) results, alloy is composed of O phase, β phase and $\alpha_2$ phase, accounting for 52%, 30% and 7% of the area, remainder is unresolved area, as shown in Figure 2. The different contents of Ti, Al and Nb elements and the solution elements lead to the variable phase composition of Ti2AlNb alloy. The β phase is BCC structure, the $\alpha_2$ phase is HCP structure, and O phase is orthorhombic structure. EBSD and EDS (energy dispersive spectroscopy) analyses are performed using a ZEISS Sigma 300 scanning electron microscope, which is equipped with an electron backscatterer and an energy spectrometer. LCF tests are carried out at 550 °C. Figure 3a shows the sample sizes of tests. LCF tests are carried out on the Instron-8802 fatigue testing machine at a constant loading rate of 0.005 $s^{-1}$ with strain control loading. When the samples fracture or the stress of the testing machine decrease by 25% instantaneously, the failure of the sample was judged. As shown in Figure 3b, the loading waveforms of LCF tests are triangular wave and the strain ratio is −1. And the Figure 3c is the evolution of maximum tensile stress per cycle. The specimens at different strain levels show slight cyclic hardening in the front and middle stages of low-cycle fatigue. At the end of the curve, there is obvious cyclic softening. The fatigue life of four strain range is shown in Table 2. When the total strain range reaches 1.6%, the fatigue life decreases to 482 cycles, the strain range can no longer increase. Therefore, the total strain ranges $\Delta\varepsilon_t$ of fatigue experiment are 1.2%, 1.3%, 1.4% and 1.6%, respectively.

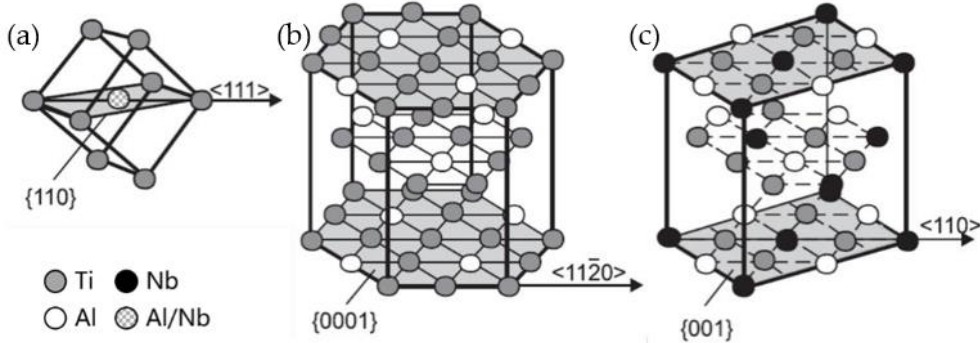

**Figure 1.** Phase structure of Ti2AlNb-based alloy: (**a**) β; (**b**) $\alpha_2$; (**c**) O [22].

**Table 1.** Chemical composition of Ti2AlNb alloy by EDS (atomic content ratio, %).

| Ti | Al | Nb | Zr |
|---|---|---|---|
| 55 | 21 | 23 | 1 |

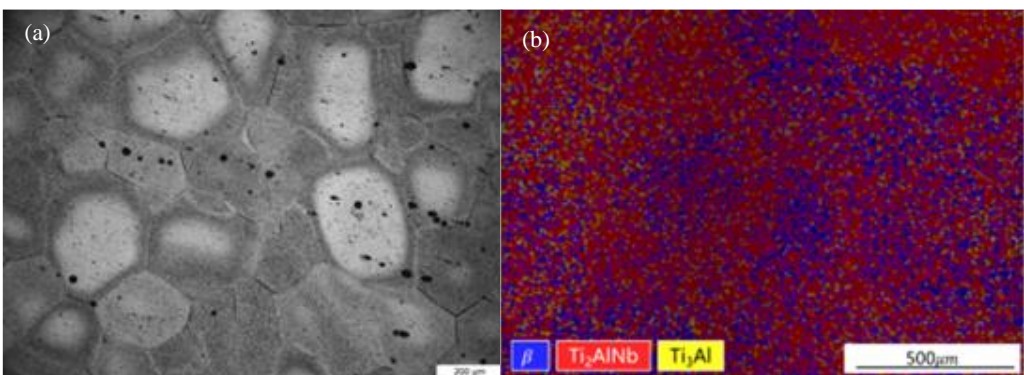

**Figure 2.** Microstructure characterization and element proportion of Ti2AlNb—base alloy: (**a**) grain morphology under OM (200×); and (**b**) phase composition under EBSD ($\beta$ is $\beta$ phase, Ti$_2$AlNb is O phase, Ti$_3$Al is $\alpha_2$ phase).

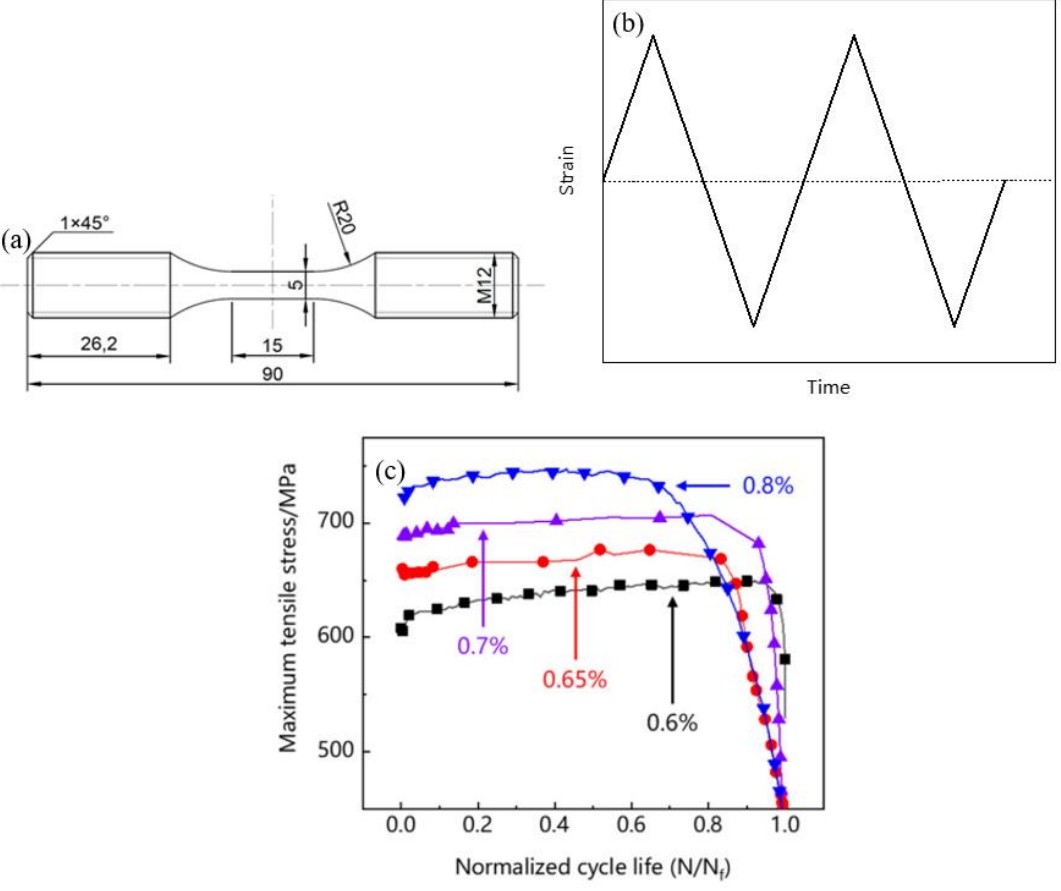

**Figure 3.** Fatigue experiments of Ti2AlNb alloy: (**a**) shape and dimensions of the LCF tests specimens; (**b**) fatigue loading waveform; and (**c**) evolution of maximum tensile stress per cycle [23].

**Table 2.** Fatigue life of four strain ranges [23].

| Test | Strain Range/% | Life |
|------|---------------|------|
| Fatigue | 1.2 | 9660/cycle |
| | 1.3 | 1081/cycle |
| | 1.4 | 742/cycle |
| | 1.6 | 482/cycle |

### 2.2. Chaboche Cyclic Deformation Constitutive Model

The Chaboche model is a time-independent model, including yield criterion, flow method, hardening criterion and strain decomposition. On the basis of the A-F follow-up hardening model, Chaboche proposed a follow-up hardening model with three back stress components called the Chaboche model [24]. The Chaboche follow-up hardening, and the Voce isotropic hardening criteria are be combined to describe the cyclic deformation behavior of Ti2AlNb base alloy at high temperature, the equations are as follows.

The total strain rate can be decomposed into elastic part and plastic part as follows:

$$\dot{\varepsilon} = \dot{\varepsilon}^{e} + \dot{\varepsilon}^{p}, \tag{1}$$

$$\sigma = \mathbf{D} : \varepsilon^{e}, \tag{2}$$

$$\dot{\varepsilon}^{p} = \frac{3}{2} \frac{\boldsymbol{\sigma}' - \boldsymbol{\alpha}}{Q} \dot{p}, \tag{3}$$

$$F = \frac{3}{2} \sqrt{(\boldsymbol{\sigma}' - \boldsymbol{\alpha}) : (\boldsymbol{\sigma}' - \boldsymbol{\alpha})} - Q^{2}, \tag{4}$$

where $\mathbf{D}$ is the elasticity tensor, $p$ is the equivalent plastic strain, $F$ is the yield function, $\alpha$ is back stress tensor and $\sigma'$ is deviatoric stress tensor. The nonlinear follow-up hardening criterion decomposes the back stress tensor $\boldsymbol{\alpha}$ into three back stress components $\alpha_i$:

$$\begin{gathered} \alpha = \sum_{i=1}^{3} \alpha_i \\ \dot{\alpha}_i = C_i \dot{p} \frac{1}{Q}(\sigma - \alpha) - \gamma_i \alpha_i \dot{p} \end{gathered} \tag{5}$$

where $C_i$ is initial value of follow-up hardening parameters, $\gamma_i$ is rate of change of following hardening parameters, $p$ is the equivalent plastic strain. The size of yield surface $Q$ is:

$$Q = Q_0 + Q_\infty(1 - e^{-bp}), \tag{6}$$

where $Q$ is the yield surface size, $Q_0$ is the initial value of the yield surface size, $Q_\infty$ is the maximum change value of the yield surface size, and $b$ is the change rate of the yield surface size.

### 2.3. Crystal Plasticity Finite Element Model

In the last century, Hill, Asaro, Rice, and Peirce [25–27] and other researchers proposed and perfected the crystal plasticity theory. The total deformation gradient $\mathbf{F}$ can be expressed by a multiplicative decomposition of elastic deformation gradient $\mathbf{F}^e$ which describes the stretching and rotation of the crystal lattice and plastic deformation gradient $\mathbf{F}^p$ associated with slip:

$$\mathbf{F} = \mathbf{F}^e \cdot \mathbf{F}^p, \tag{7}$$

Due to the plastic deformation is the result of slip, the plastic component of velocity gradient $\mathbf{L}^p$ can be written as the sum of slip rate on all activated slip systems:

$$\mathbf{L}^p = \dot{\mathbf{F}}^p \mathbf{F}^{p-1} = \sum_{\alpha=1}^{n} \dot{\gamma}^{\alpha} \mathbf{m}^{\alpha} \otimes \mathbf{n}^{\alpha}, \tag{8}$$

where $\dot{\gamma}^{\alpha}$ is the slip rate on the $\alpha$th slip system. $\mathbf{m}^{\alpha}$ and $\mathbf{n}^{\alpha}$ are the slip direction and slip plane normal of the $\alpha$th slip system, respectively. In the current work, in order to describe the crystal plasticity under cycle loading and the effect of temperature on rate-dependent deformation behavior, a thermally activated flow rule [28,29], with $\dot{\gamma}^{\alpha}$ as a function of $\tau^{\alpha}$, $B^{\alpha}$ and $S^{\alpha}$, is adopted as follows:

$$\dot{\gamma}^{\alpha} = \dot{\gamma}_0 exp\left[-\frac{F_0}{k\theta}\left\langle 1 - \left\langle \frac{|\tau^{\alpha} - B^{\alpha}| - S^{\alpha}}{\tau_0} \right\rangle \right\rangle\right] sgn(\tau^{\alpha} - B^{\alpha}), \tag{9}$$

where $\dot{\gamma}^{\alpha}$, $k$, $F_0$ and $\theta$ are the reference strain rate, Boltzmann constant, Helmholtz free energy for material and absolute temperature, respectively. $\tau^{\alpha}$, $B^{\alpha}$ and $S^{\alpha}$ represents the resolved shear stress, back stress, and isotropic slip resistance on the $\alpha$th slip system. $\tau_0$ is the critical slip resistance at the current temperature, and function $<x>$ indicate that:

$$\langle x \rangle = \begin{cases} x(x \geq 0) \\ 0(x < 0) \end{cases}, \tag{10}$$

The slip resistance $S^{\alpha}$ is defined as

$$\dot{S}^{\alpha} = \sum_{\beta=1}^{N} h^{\alpha\beta}\left(\frac{S_{sat} - S^{\beta}}{S_{sat} - S_0}\right)\left|\dot{\gamma}^{\beta}\right|, \tag{11}$$

where $h^{\alpha\beta}$ is the hardening matrix to indicate the cross-hardening behavior between the slip systems $\alpha$ and $\beta$. $S_{sat}$ and $S_0$ are the saturated and initial slip resistance, respectively. The evolution of the back stress $B^{\alpha}$ follows Armstrong–Frederick's kinematic hardening rule [23,30,31]:

$$\dot{B}^{\alpha} = h_B\dot{\gamma}^{\alpha} - \frac{r_D}{S^{\alpha}}B^{\alpha}\left|\dot{\gamma}^{\alpha}\right|, \tag{12}$$

where $h_B$ are $r_D$ material constants.

## 3. Model Parameters Identification and Validation

### 3.1. Chaoboche Cyclic Deformation Constitutive Model

The finite element model was established by taking a cube within the gauge of the cylindrical sample. The model is shown in Figure 4. The constraints are applied to the bottom of the Y direction and Z direction. In this paper, the Chaboche model was used to simulate macroscopic deformation. Ti2AlNb alloy uniaxial tensile test parameters were automatically fitted by the method in the ABAQUS user manual, and the fitting results are shown in Table 3. The model parameters in Table 3 were used to simulate the LCF stress–strain response of Ti2AlNb alloy at 550 °C, and the results are shown in Figure 5. Due to the hysteresis loop reaching the stable stage after 10 cycles, the model parameters were adjusted based on the steady-state hysteresis loop. It can be seen that the simulation results fit with the test results.

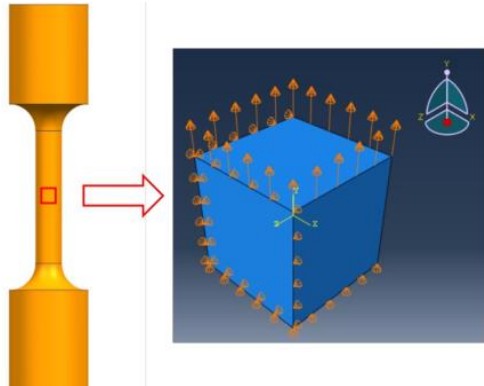

**Figure 4.** Element model and boundary condition.

**Table 3.** Fitting values of elastic and kinematic hardening parameters.

| E/GPa | $Q_0$/MPa | $C_1$/MPa | $\gamma_1$ | $C_2$/MPa | $\gamma_2$ | $C_3$/MPa | $\gamma_3$ |
|---|---|---|---|---|---|---|---|
| 109.191 | 570 | 1,634,660 | 25,468 | 142,294 | 1405.3 | 36,262 | 187.15 |

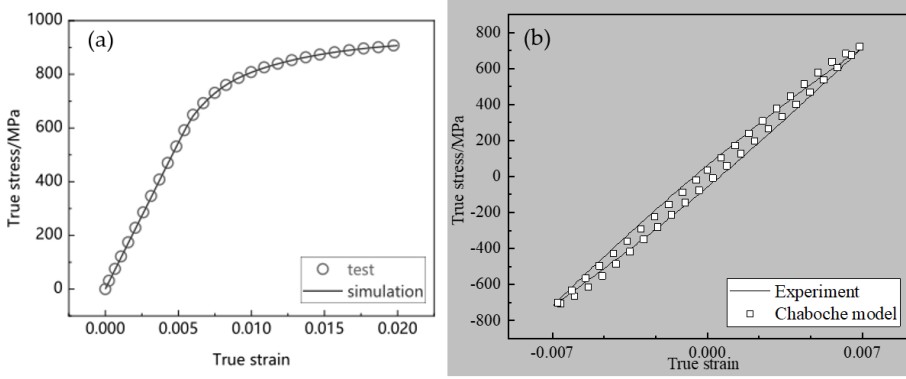

**Figure 5.** Simulation, (**a**) and experimental results of uniaxial tension and LCF steady-state hysteresis loop at 1.4% strain range, (**b**).

### 3.2. Crystal Plasticity Finite Element Model

#### 3.2.1. RVEs and Boundary Conditions

Based on the EBSD phase composition figure, the 3D RVEs containing two phases were established as shown in Figure 6. According to EBSD results, O phase, β phase and $\alpha_2$ phase accounted for 52%, 30% and 7%, respectively. Due to the least content of $\alpha_2$ phase, this was ignored, so the $Ti_2AlNb$ was regarded as a two-phase material. The RVE contained 100 sets, the proportion of O phase and β phase was 2:1. Figure 6a shows that the O phase is the dominant phase, and the β phase is dispersed in the O phase. The RVEs were generated by Neper [32], the mesh scale was 100 × 100, and were divided into 100 areas. Then the RVE files were imported into ABAQUS. The boundary conditions for the RVE model are shown in Figure 6, the periodic boundary conditions (PBC) were implemented, and can be expressed as:

$$u(CD) - u(AB) = u(C) - u(A), \tag{13}$$

$$v(BD) - v(AC) = v(B) - v(A), \tag{14}$$

where u(AB), u(CD), v(BD) and v(AC) are the x-direction and y-direction displacements for edges AB, CD, BD and AC. u(A), u(C), v(B) and v(A) are the x and y displacements of nodes A, B and C. To remove the rigid body motion, the vertex A was fixed in both x and y directions, and B and C were fixed in the x and y direction, respectively. A displacement was applied to vertex C in the x-direction to simulate the fatigue loading condition.

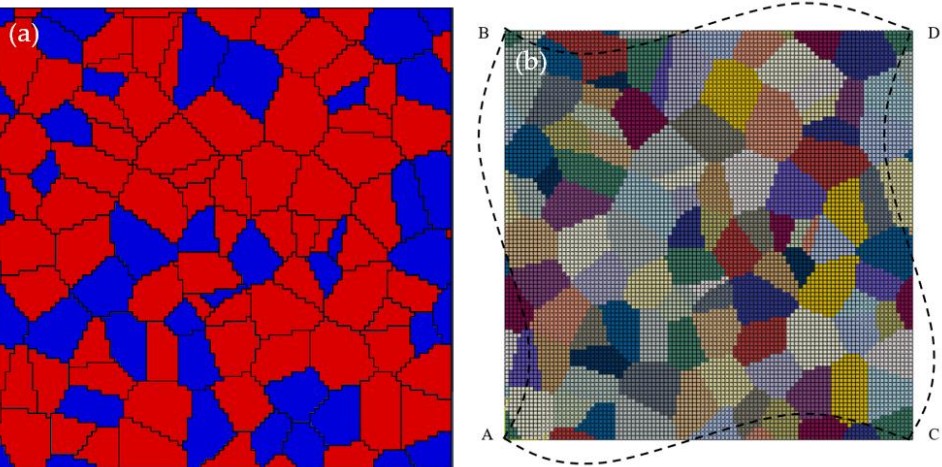

**Figure 6.** (**a**) RVE with O phase (red) and β phase (blue); and (**b**) periodic boundary conditions.

### 3.2.2. Identification of Material Parameters

The material parameters of O phase and β phase used in current CPFE model are shown in Table 4, respectively. The elasticity constants are cited from Chu [33], Fan [34] and Fu [35]. However, the elasticity constants of Ti2AlNb alloy at 550 °C have not been reported so far. Therefore, in this paper, the elasticity constants at 550 °C are obtained by revising the values at room temperature. According to the previous studies, the crystal structure of O phase is orthorhombic structure with cmcm symmetry. At room temperature the activated slip systems are <100> {001} and <110> {110}, and the slip resistance of <110> {110} is greater than <100> {001} [36]. When loading temperature is 550 °C, more slip systems are activated, but new slip systems are not all activated through whole cycle deformation. Therefore, contributions from other slip systems will be added to <100> {001} and <110> {110}. The crystal system of β phase is BCC, so the slip system of β phase is <111> {110}.

**Table 4.** Material parameters of the crystal plasticity-based formulations for Ti2AlNb.

| | Parameter | Unit | Value(O | β) | |
|---|---|---|---|---|
| Elastic constants | $C_{11}$ | GPa | 184 | 135 |
| | $C_{12}$ | GPa | 86.2 | 113 |
| | $C_{44}$ | GPa | 49 | 55 |
| Flow parameters | $\dot{\gamma}_0$ | $s^{-1}$ | 120 | 120 |
| | $F_0$ | kJ/mol | 150 | 250 |
| | $\tau_0$ | MPa | 200 | 200 |
| Hardening parameters | $h_B$ | MPa | 1000 | 950 |
| | $r_D$ | MPa | 8 | 10 |
| | $S_{sat}$ | MPa | 250 | 160 |
| | $S_0$ | MPa | 300 | 225 |
| | $h^{\alpha\beta}$ | MPa | 350 | 350 |

The parameter fitting procedures are shown as follows. First, the parameters at room temperature are used as the initial data and the approximate range of each parameter was obtained. Then, the parameters were adjusted to fit the uniaxial tension simulation and experimental curves. Finally, a cyclic loading simulation was performed using the parameter set of uniaxial tension. Due to the hysteresis loop reaching the stable stage after 10 cycles, the model parameters were adjusted based on the steady-state hysteresis loop. The true stress–strain curve is shown in Figure 7. It can be seen that the simulation results fit with the test results.

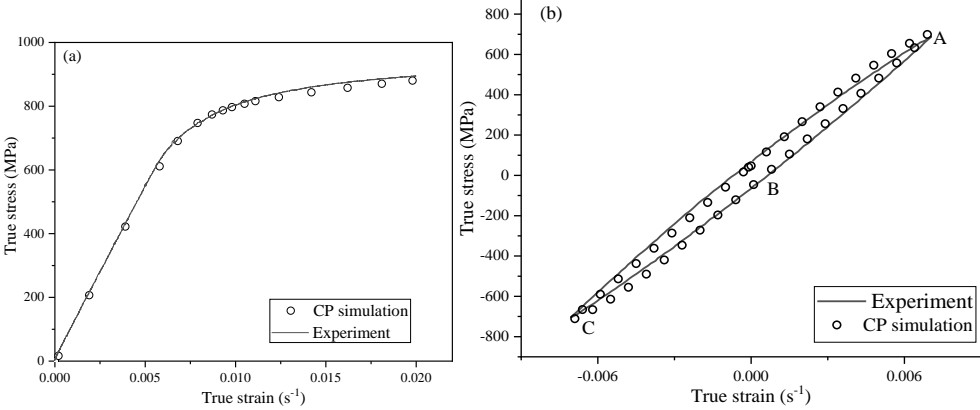

**Figure 7.** Crystal plasticity simulation and experimental results: (**a**) 2% uniaxial tension; and (**b**) steady-state hysteresis loop at 1.4% cycle deformation.

## 4. Results and Discussion

### 4.1. Comparison of Simulation Ability between Cyclic Deformation Model and CPFEM

Figure 8 shows the true stress–strain curves of experiment, CP simulation and Chaboche model at 1.2%, 1.3%, 1.4% and 1.6% four strain range. From the comparison figure, the CP simulation and Chaboche model fit well with the experiment at 1.2%, 1.3% and 1.4% strain range. This indicated that the simulation accuracy of CPFEM can reach the same level as the macroscopic constitutive model. However, at 1.6% strain range, the stress of CP simulation and Chaboche model in the tension and compression stage was slightly higher than the experiment, which may have been the result of a calculation error. From the above simulation results, a 1.4% strain range was determined to be the best, so the following crystal plasticity analysis used the 1.4% strain range as an example.

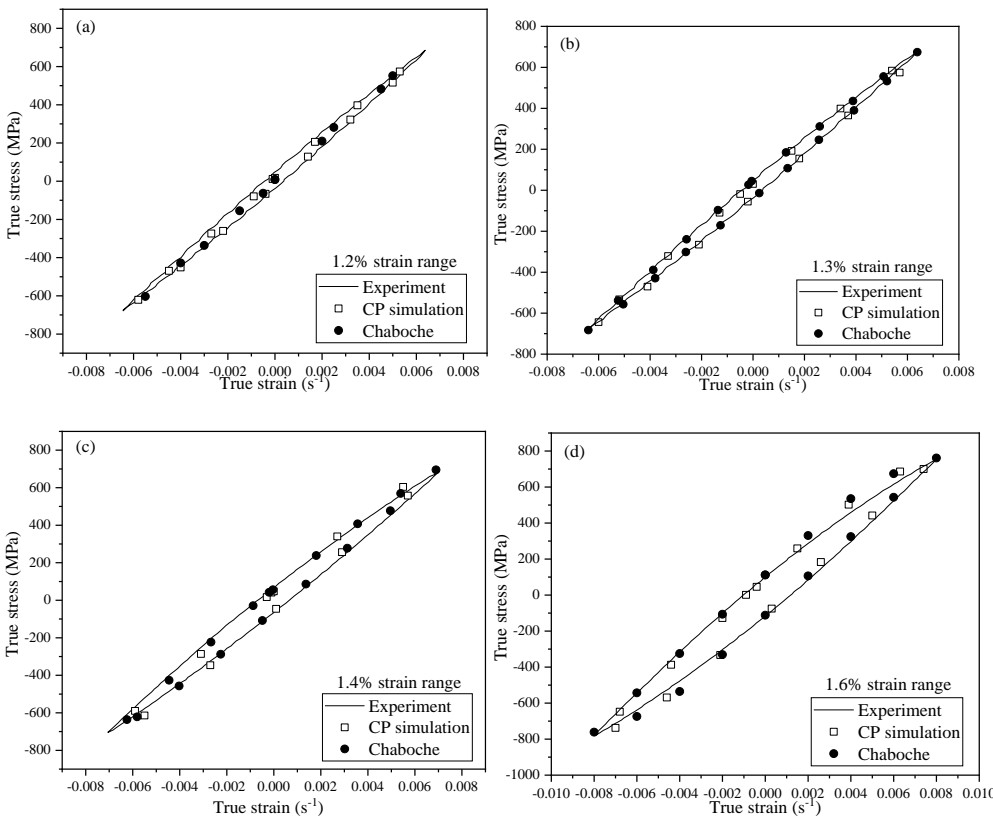

**Figure 8.** Stress–strain curves of experiment, CP simulation and Chaboche model: (**a**) 1.2% strain range; (**b**) 1.3% strain range; (**c**) 1.4% strain range; and (**d**) 1.6% strain range.

### 4.2. Strain and Stress Distribution between the O Phase and b Phase

For the Ti2AlNb-based alloy, the crystal system and activated slip systems were different between the O phase and $\beta$ phase. This will result in an incoordination in deformation process of two phases and may cause stress concentration at the phase boundary. Therefore, in this paper, CPFEM was used to obtain the stress and strain distribution figures of two phases in different cycle stages. The simulation results showed that fatigue reached a stable stage after 10 times cycle deformation, and following studies are based on these results.

At the 10th cycle loading, the stress and strain distribution figures of three loading points that 0.7% strain (maximum tension point, point A in Figure 7), 0 strain (zero point, point B in Figure 7) and −0.7% strain (maximum compression point, point C in Figure 7) were selected, as shown in Figure 9. The Figure 9a,e is stress distribution of ±0.7% strain, the points where the stress is significantly higher are highlighted by red circle. All the stress concentration points are located at the phase boundary, and the points at compression stage are greater than the points at the tension stage. Figure 9c shows the stress distribution of B

point, even though the strain is 0, the stress at the boundary still reaches up to 300 MPa. Figure 9d,f are the strain distribution of point A, B and C, respectively. There is a large difference between the strain of the O phase and that of the β phase. The strain of the O phase was 0.2–0.8%, the strain of β phase was much greater than 1%, suggesting m the sufficient plastic deformation of the β phase. From the research of Wen [37] and Li [38], O phase as the hardening phase will lead to dislocation accumulation, and high-density dislocations limit the deformation of the β phase.

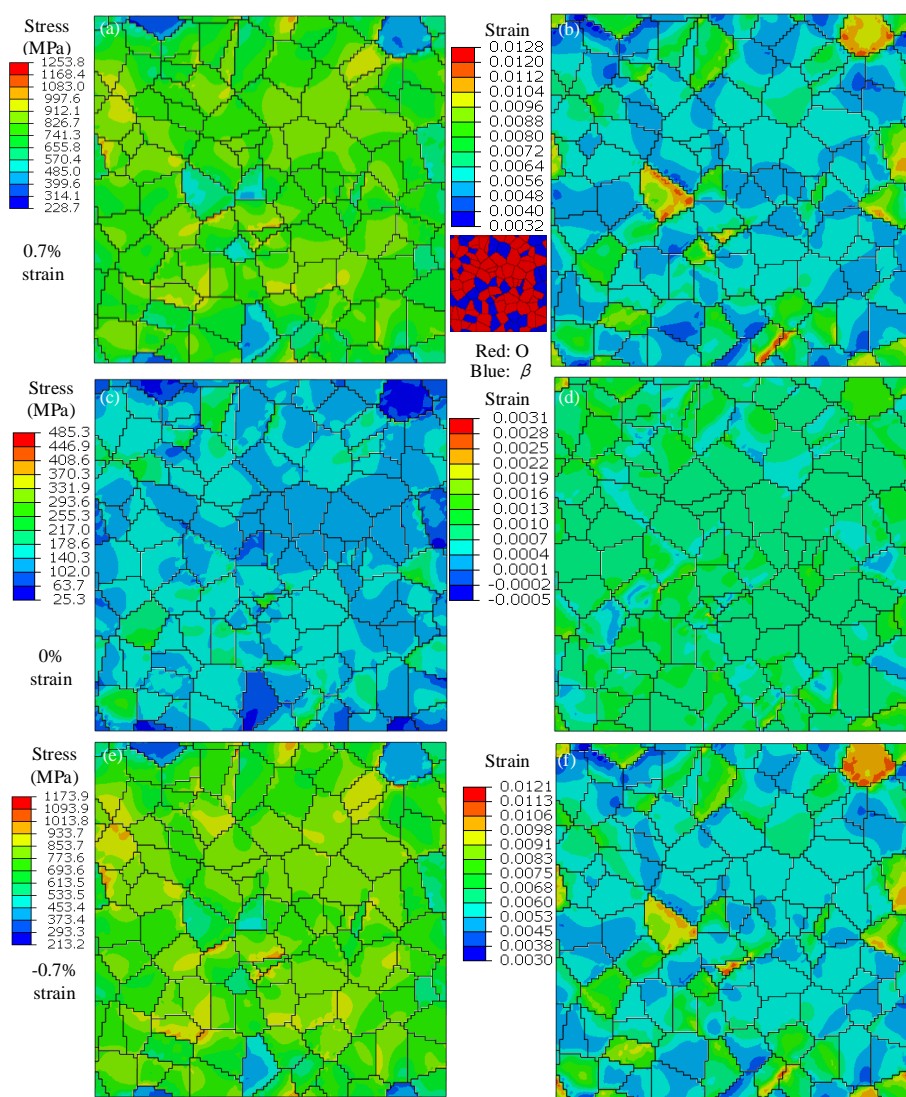

**Figure 9.** Stress and strain distribution figures of three loading points: (**a**,**b**) 0.7% strain; (**c**,**d**) 0% strain; (**e**,**f**) −0.7% strain. (The O and β phases are also shown in figure, as same as Figure 6a).

According to the discussion of strain and stress distribution, it was found that phase boundaries are the most important positions relating to stress concentration. Figure 10 shows the rotation angle and stress at 0.7% strain. The rotation angle distribution of each grain is shown in Figure 10a, the stress distribution is shown in Figure 10b. Due to the total strain range at 1.4%, the rotation angles of each grain during fatigue loading are 0.22~0.62°. Although the grain rotation is not significant, the grain rotation angle between the two phases is obviously compared, and the grain rotation has a strong correlation with the stress concentration. The rotation angle is closely related to the phase, and most of the grains with a rotation angle larger than 0.4° are β phase, and most of them are stress concertation positions, as shown in Figure 10b (circled by solid and dotted lines). The

dotted line circles show the position of stress concentration at the grain with large rotation angle, and the solid line circles show the position of no stress concentration at the grain with a large rotation angle. The dotted line circles all are O/β phase boundary, and the solid line circles are all same phase boundary. The plastic deformation ability of the two phases is different; during fatigue loading, β phase is more prone to plastic deformation and grain rotation is more significant. The strain incompatibility between the two phases is more serious, which makes the O/B boundary prone to stress concentration.

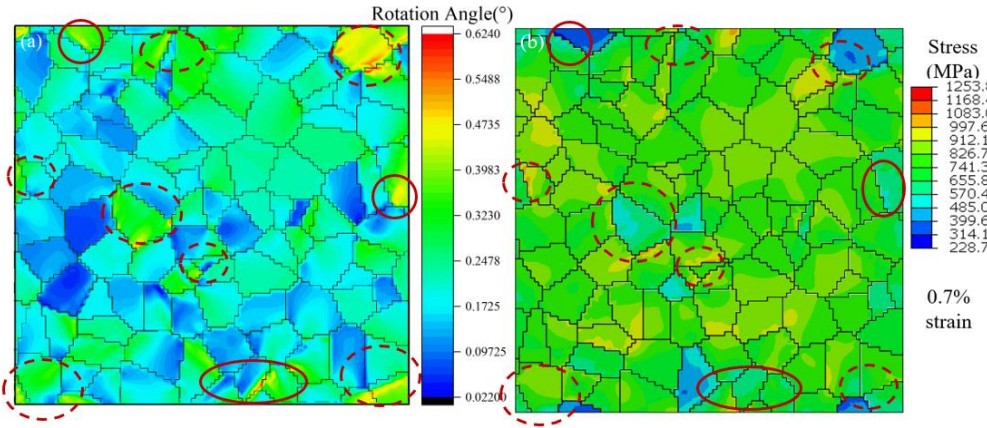

**Figure 10.** 0.7% strain: (**a**) the grain rotation angle distribution; and (**b**) the stress distribution.

## 5. Conclusions

In this work, a CPFE model based on EBSD and EDS images was conducted to study the grain-level stress and strain distribution under cycle loading of Ti2AlNb alloy. The following conclusions were obtained:

1. The simulation performance of CPFEM is the same as that of Chaboche model, but the deformation mechanism Ti2AlNb alloy at high temperature is complex, so with the larger strain range, simulation performance deteriorates;

2. The stress concentration at the tension and compression stage is caused by the deformability of the O and β phases. The O phase is harder than the β phase, and the grain rotates at a smaller angle and has less strain. The inhomogeneous deformation at the phase boundary led to stress concentration.

**Author Contributions:** Conceptualization, H.L.; Data curation, X.L.; Formal analysis, Y.W., Z.Z. and Y.Y.; Investigation, Y.W. and X.W.; Methodology, H.L.; Project administration, H.L.; Resources, Y.W.; Software, Y.W. and Z.Z.; Validation, Z.Z.; Visualization, Z.Z. All authors have read and agreed to the published version of the manuscript.

**Funding:** This research was funded by National Major Science and Technology Project (J2009-VI-0003-0116). The authors also gratefully acknowledge the support of the National Natural Science Foundation of China with (51835011 and 51775441).

**Institutional Review Board Statement:** Not applicable.

**Informed Consent Statement:** Not applicable.

**Data Availability Statement:** Due to the lack of unanimous consent from all the authors. Data is not shared.

**Acknowledgments:** We thank Tingzhuang Han, Xiaowei Yi and Gang Ran for their help in the experiments and discussion.

**Conflicts of Interest:** The authors declare that the work is original research that has not been published previously, and is not under consideration for publication elsewhere. No conflict of interest exist in the submission of this paper, and the paper is approved by all authors for publication.

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
