# Peer review of "Crystal Plasticity Finite Element Modeling on High Temperature Low Cycle Fatigue of Ti2AlNb Alloy"

_applsci, doi:10.3390/app13020706_

Round 1

Reviewer 1 Report

Dear Editor,

This article is about the cyclic fatigue simulation of a titanium alloy using the crystal plasticity finite element method. The manuscript itself is well written but there is a complete lack of data analysis and several flaws in this study. Therefore I recommend to reject the article in the current form.

Major comments are as follows:

1) This is not a "superalloy" as the mechanical properties are pretty standard for a titanium alloy and this will have no high temperature resistance.

2) The introduction is confusing, you mix nickel and titanium alloys, all the reasoning is no sense, titanium and nickel have different applications in a jet engine. It's not one is better than the other.

3) Section 2.2 must be improved, some symbols have no definition like "p"

4) The main criticism I have of your calibration is that the data you use do not allow you to reliably determine the backstress evolution coefficients because your strain amplitude is not large enough. This is demonstrated by your very different coefficients C_1, C_2 and C_3. Most likely only one backstress component is dominating and your Chaboche model will not be able to fit cyclic data at higher strain amplitude. Either you simplify the model or you calibrate with larger strain amplitude data. In general your value of C_1 is too large, normally in a Friedrick-Angstrom model that coefficient is of the order of 10^4 MPa for metals. There must be an error here.

5) The symbol S in equations (6) and (7) is used for different quantities.

6) Since there is too little plasticity, from figure 5, it is very difficult to deduce if the model is fitting well the data or not. You need to show how the simulated curves change with the value of C_1

7) It is not very clear what is the purpose of comparing crystal plasticity with Chaboche because this comparison is never discussed or used afterwards.

8) You mention several experiments in reference [10], [11], [12] but there is no discussion in this paper, therefore you don't compare your simulation results to understand what is consistent with experiments and what not.

9) There is no mention about the crystalline structures of the two phases

10) The author jumps too quickly to the conclusion that the disagreement of the stress at 1.4% amplitude is due to some softening mechanism, while most likely it is a problem with the calibration. Indeed it seems no systematic or automated calibration procedure has been used, therefore errors are likely.

11) The author starts discussing about fatigue damage but there is no model for fatigue damage here used.

12) After figure 9 I would have expected some discussion about why the stress concentration takes place in some specific grains, but the article suddenly finish without an explanation. You have to analyze the state variables and grain orientation and phase properties and give a clear argument about why specific grain combinations show high stress.

Best Regards

Reviewer 2 Report

This is a nice piece of work discussing the possible cause of low-cycle fatigue at high temperatures using a crystal-plasticity finite element model. The main conclusion is that fatigue is caused by residual stresses in the phase boundaries. I think this work is publishable after some technical details are supplemented.

1.     Since the finite element model is a RVE, it is unclear how the periodic boundary conditions are applied in the ABAQUS. If there is an additional code, please provide it.

2.     Although the fatigue experiments were conducted, the fatigue results in terms of S-N curve were not provided.

3.     I do not see the fatigue result (e.g., S-N curve or a tendency of failure) based on the finite element model as well. Has a failure criterion been imposed in the model?

4.     The authors indicate that the parameters at high temperatures are unavailable and that room-temperature parameters can be found. However, they did not check their model by comparing simulation results with room-temperature experimental results.

5.     Finally, where is the temperature effect?

Reviewer 3 Report

The English writing is very poor. It is difficult to read through because there are too many English mistakes. For example, only in the Abstract section, I have found 5 mistakes:

Line 14: “two phase” should be “two phases”

Line 15: “fatigue experiment” should be “fatigue experiments”

Line 15: “Chaboche model” should be “the Chaboche model”

Line 17: “important positions” should be “important position”

Line 17: “may causes” should be “may cause”

The authors should use English editing service to improve the manuscript if they cannot do it by themselves. Besides, the writing is hard to understand.

In section 2.1, it is mentioned that the fatigue test was conducted at a constant loading rate of 0.005 s^-1. Do you use position-control mode or strain-control mode? If the deformation is done by position control to control crosshead speed, then your loading rate is a nominal rate. If you use strain control to control the sample strain that is measured by an extensometer, then the loading rate is a real rate. Please explain this with more details in section 2.1.

For all the equations in the whole manuscript, the authors mix up vectors, tensors, and scalars. Usually, vectors and tensors are written in bold italic symbols, while scalars are written in italic symbols. However, all the symbols in this manuscript are written in italic symbols, which is not acceptable.

Eq. (3) is not well explained; please explain why the subscript I vary from 1 to 3. Usually, the back stress component is a stress tensor, but yours is not.

In equation 6, you used S^α to represent the slip direction. However, in equations 7 and 9, you used S^α again to represent isotropic deformation resistance.

The manuscript title is about low cycle fatigue, but you only simulated one deformation cycle. The experimental data is also from only one deformation cycle.

In the conclusion section, you mentioned the fast propagation of microcracks. However, in this manuscript, you didn’t simulate damage or crack propagation. Why did you put speculation into the conclusions section?

Overall, this reviewer suggests a rejection to publish it in this journal because of the above problems.

Reviewer 4 Report

Experimental part.

The authors do not specify what the acronyms EBSD and EDS stand for. The first time they appear in the text they should be specified.

The authors do not specify which microscope or equipment they use to do EDS or EDX.

Why don't the authors microanalyse the sample and do EDS?

How do the authors know the percentage of each phase in the alloy without doing XRD?

The authors do not specify whether the alloy is commercial or how they arrive at the alloy. It is of utmost importance to justify this in the text as this information is relevant to the research.

In figure 8 the symbols used in the graph are not distinguishable, they should be modified so that they can be visually observed.

In figure 9 the legends of the graphs cannot be read correctly, so they should be enlarged as much as possible.

Round 2

Reviewer 1 Report

Dear Authors,

the manuscript has improved but you need to put the effort to do an automated calibration of the stress strain curves in figure 8 or you will never be able to find accurate coefficients for the Chaboche model.

Also, the analysis of the grain pairs in figure 10 must be automated and you need to analyze all grain pairs, otherwise there is not enough statistics for reaching conclusions.

Also, your literature review seems to be carried out time ago and does not contain recent publications about crystal plasticity of titanium alloys, such as:

https://www.sciencedirect.com/science/article/pii/S2238785422011334

and

https://www.sciencedirect.com/science/article/pii/S1005030222004881

Reviewer 2 Report

I think the responses are inappropriate. 

Q1. If the simulation model is an RVE--this term means it can be repeated to form the macroscopic region--then how can you ignore periodic boundary conditions? I think you simulated the tensile test of a block and believed that the behavior of the block could represent the behavior of a macroscopic experimental sample. This treatment is fine, but the model is no longer an RVE.

Q2. Without S-N curve there is no information about fatigue. It is simply loading-unloading tests. You shall consider changing the title of the manuscript.

Q3. Similarly, what kind of tests have been done? Fatigue test or loading-unloading test? How can you use one or several cycles without failure to represent fatigue property? Without a failure criterion, how to predict fatigue?

Q4. Using parameters from a reference does not guarantee that your model corresponds to your specimen. For example, your material can be different from the one in reference in chemical composition, phase constituent, phase morphology, grain size, etc. They could be similar, but you shall have proof.

Q5. I do not mean the equation. I mean whether the temperature effect from the simulation model can be consistent with the temperature effect from experiments.

Reviewer 3 Report

The authors have improved the paper according to the comments. Now it can be accepted.

Author Response

  • Thank you very much for your recognition of our work.

Round 3

Reviewer 2 Report

Thanks for the effort. I have no more comments.

Author Response

Thank you very much for your recognition of our work.